# Adverse Childhood Experiences and the Risk of Multiple Sclerosis Development: A Review of Potential Mechanisms

**DOI:** 10.3390/ijms25031520

**Published:** 2024-01-26

**Authors:** Karine Eid, Marte-Helene Bjørk, Nils Erik Gilhus, Øivind Torkildsen

**Affiliations:** 1Department of Neurology, Haukeland University Hospital, Jonas Lies vei 71, 5053 Bergen, Norway; marte.bjork@uib.no (M.-H.B.); nils.gilhus@uib.no (N.E.G.); 2Department of Clinical Medicine, University of Bergen, 5021 Bergen, Norway; oivind.fredvik.grytten.torkildsen@helse-bergen.no; 3NorHead, Norwegian Center for Headache Research, 5021 Bergen, Norway; 4Neuro-SysMed, Department of Neurology, Haukeland University Hospital, 5021 Bergen, Norway

**Keywords:** childhood, early, stress, trauma, violence, abuse, MS

## Abstract

Adverse childhood experiences (ACEs), such as abuse, neglect, and household dysfunction, contribute to long-term systemic toxic stress and inflammation that may last well into adulthood. Such early-life stressors have been associated with increased susceptibility to multiple sclerosis (MS) in observational studies and with the development of experimental autoimmune encephalomyelitis in animal models. In this review, we summarize the evidence for an ACE-mediated increase in MS risk, as well as the potential mechanisms for this association. ACEs dysregulate neurodevelopment, stress responses, and immune reactivity; they also alter the interplay between the immune system and neural networks. All of this may be relevant for MS risk. We further discuss how ACEs induce epigenetic changes and how the toxic stress caused by ACEs may reactivate the Epstein-Barr Virus (EBV), a key risk factor for MS. We conclude by suggesting new initiatives to obtain further insights into this topic.

## 1. Introduction

Multiple sclerosis (MS) is a chronic inflammatory and neurodegenerative disease affecting the central nervous system (CNS) caused by an interplay between genetic and environmental factors. Epstein-Barr Virus (EBV) infection is hypothesized to be a causal risk factor for MS and is associated with a 32-fold increase in MS risk [1]. Studies have found that EBV infection in adolescence is associated with a particularly increased risk [2,3] suggesting that adolescence is a critical period for MS susceptibility in adult-onset MS. EBV infection is probably not sufficient to cause MS by itself. Several other environmental risk factors have shown consistent associations with increased MS risk, such as vitamin D deficiency, low sun exposure, a history of smoking, and a high body mass index [4,5] Exposure at a young age, especially in adolescence, seems to be critical for these risk factors [6,7,8,9,10].

Psychological stress has been considered to affect MS since the initial description of the disease in the 19th century, when Jean-Martin Charcot described “long-continued grief or vexation” as being related to the onset of the disease [11]. However, stress is still a controversial risk factor for MS due to limited evidence and heterogeneous study designs in the existing literature [12,13] The association between stress and MS disease activity has been studied more thoroughly than the association between stress and MS susceptibility [12]. Stress has been associated with MS relapse in both prospective and case-control studies, and the risk of MS relapse is especially elevated during the first months after a stressful event [12]. In this review, we will focus on the association between stress and MS susceptibility.

Some studies have found that stressful events, such as divorce, interpersonal conflicts, and severe illness, happen more often among people with MS than among controls 1–5 years before MS symptom onset or diagnosis [14,15]. A prodromal phase of MS can, in retrospect, sometimes be identified many years before the first evident MS symptoms [16]. Neurodegeneration and inflammation may therefore start long before MS diagnosis [17,18]. The finding that people with MS experience more stressful events close to clinical MS onset compared to controls may illustrate that stress in the prodromal phase could boost ongoing pathological mechanisms to reach a clinical threshold. It remains unknown how long prior to MS symptom onset one should investigate risk exposure to assess causal factors for MS. However, childhood and adolescence have repeatedly been demonstrated to be time periods of particular interest.

Both epidemiological and experimental studies have found an association between adverse childhood experiences (ACEs) and MS [19,20,21,22,23]. ACEs represent potentially traumatic events, such as abuse or neglect, witnessing violence, or growing up in a household with drug use, mental health problems, or instability due to parental separation, death, or incarceration. Such early-life stressors are known to have long-term negative consequences for lifestyle choices and health in adulthood [24,25]. The Kaiser Permanente ACE Study was the first large cohort study to demonstrate an association between childhood abuse, neglect, and household dysfunction and negative health outcomes decades later [24]. The study showed that ACEs increase the risk of adult chronic diseases, such as autoimmune disorders, cancer, cardiovascular disorders, and psychiatric disorders. An underlying hypothesis is that ACEs initiate a “toxic stress response”, a prolonged activation and dysregulation of biological stress systems, which negatively influences neuro-immune-endocrine pathways and the interplay between them [26,27]. The imprint of childhood trauma on such biological pathways may persist over the lifespan [27].

In this review, we will summarize the current evidence of how ACEs increase the risk of adult-onset MS from epidemiological and experimental studies and discuss the underlying biological mechanisms.

## 2. Search Strategy

To identify research papers investigating the association between ACEs and MS, we performed an unstructured search for peer-reviewed English literature listed in PubMed using the search terms childhood trauma, childhood stress, adverse childhood experiences, adverse childhood events, early-life trauma, and early-life stress, combined with multiple sclerosis. We started the search on 24 November and ended it on 1 December 2023. We searched for all the papers ever published on this topic. We first screened for relevant titles and subsequently read the abstracts. We reviewed the full text of all eligible abstracts where full text was available through either open access or subscription by the University Library, University of Bergen. Papers were excluded if the number of MS cases was not reported or was <40, or if there were no comparison groups. We also used a backward and forward snowballing approach to search for additional papers not identified by the original search. 

## 3. Adverse Childhood Experiences Are Associated with Increased MS Risk

Both experimental and epidemiological studies have investigated the association between MS risk and early stress or ACEs, and the majority of studies were published after 2010.

### 3.1. Experimental Studies

An association between early-life stress and the development of MS was supported by an experimental study that triggered early-life emotional and physical trauma in mice and found increased susceptibility to experimental autoimmune encephalomyelitis (EAE) [20]. The EAE tended to be more severe and more resistant to interferon-β treatment than in the control mice. Neonatal mice were separated from their mothers and injected with saline to induce emotional and physical distress. Exposed mice had downregulated adrenergic receptors on innate immune cells, which may result in a reduced capacity to respond adequately to stress and inflammation.

Another study used shipment of neonatal laboratory mice as an early-life stressor and found that exposed mice had increased risk of EAE after immunization. They also suffered more severe clinical symptoms and had less recovery than unexposed mice [22]. The researchers found higher circulating levels of stress hormones in adult EAE mice exposed to early-life stress compared to unexposed EAE mice with the same immunization.

A study using Theiler’s murine encephalomyelitis virus model found that adolescent mice exposed to prolonged maternal separation in early life had higher virus antibody titers after infection that also persisted longer, compared to control mice and mice exposed to only brief maternal separation [23]. Additionally, mice exposed to neonatal stress showed inadequate release of stress hormones after viral infection. This indicates that early-life stressors can disrupt the hypothalamus-adrenal axis and the response of the innate immune to a viral infection.

### 3.2. Epidemiological Studies

We identified 10 epidemiological studies investigating the association between ACEs and MS risk in our literature search. Five of these studies were case-control studies, one was a cross-sectional study, and three were cohort studies. An overview of these epidemiological studies and their main findings is shown in Table 1.

### 3.3. Case-Control and Cross-Sectional Studies

An association between ACEs and MS was reported in a German case-control study of 234 people with MS (Table 1) [36]. The study investigated different categories of abuse and neglect and found an increased risk of having experienced sexual or emotional abuse in childhood, as well as emotional neglect, among those with MS compared to controls [36]. A case-control study from Iran with 250 MS cases found an increased risk of exposure to weekly physical abuse in childhood compared to controls, with an unadjusted odds ratio of 18.8 [34]. A Canadian case-control study among people with immune-mediated inflammatory diseases, including 232 individuals with MS, found an increased risk of childhood maltreatment among those with disease compared to healthy controls [31], with an odds ratio of 2.4 for emotional abuse. However, a case-control study from California including 1422 MS cases did not find any increased risk of reporting ACEs in telephone interviews [29]. The ACEs included parental divorce, death or illness in the core family, disruption in living situation, and childhood abuse (physical and verbal combined). A cross-sectional study from the Icelandic Stress-And-Gene-Analysis cohort of 28,000 women, including 214 women with MS, investigated 13 ACEs [30]. The authors did not find any significant associations with MS. They found elevated risk estimates for bullying, physical neglect, parental separation, and severe sexual abuse, but the confidence intervals were wide and contained the null after adjustment for confounders.

Other, smaller studies (<100 MS cases) found both increased risk [33] and no increased risk [32] of self-reported abuse and neglect in childhood in people with MS.

### 3.4. Cohort Studies

A population-based Danish cohort study with 2,973,993 people, including 3260 individuals with MS, assessed early-life stressors recorded in national registries, such as parental divorce and the death of parents or siblings. They found that exposure to a stressful event before the age of 18 years was associated with an 11% increased risk of MS (Table 1) [35]. The association was mainly driven by exposure to parental divorce. Experiencing the loss of a parent or sibling did not increase the risk of MS.

A cohort study with 116,671 female participants that included 262 women with MS from the Nurses’ Health Study did not find that exposure to physical or sexual abuse in childhood or adolescence increased the risk of MS [37]. The study used a mixed cohort design. Childhood abuse was mainly assessed retrospectively. The study also included a subgroup of 49 women who developed MS 1–4 years after responding to the questionnaire regarding exposure to childhood abuse.

In a large Norwegian cohort study with a prospective design, the authors followed 77,997 women who answered questionnaires regarding different types of abuse [28]. A total of 300 women developed MS during a median of 7 years of followup (range 0–17 years). The hazard rate for developing MS was 31% higher for those who had experienced any type of sexual, emotional, or physical abuse before the age of 18 years. The risk of MS was highest after sexual abuse, followed by emotional abuse, and the study found a dose–response relationship between the number of abuse categories and the risk of MS [28].

### 3.5. Systematic Reviews

Two systematic reviews have summarized the evidence from observational studies investigating the association between ACEs and MS [19,21]. The systematic reviews investigated MS onset and other clinical features of MS in association with ACEs. Both reviews concluded that the evidence supports a link between childhood stress and MS risk. They also reported that people with a history of ACEs develop MS symptoms at a younger age, that they have more fatigue and exaggerated reaction to pain, and that ACE severity is associated with an increased rate of MS relapses [19,21].

### 3.6. ACEs and MS Risk: Summary of Evidence

Taken together, the current evidence supports an association between ACEs and MS susceptibility. Among the 10 included epidemiological studies on childhood stress and MS risk, six found an association with MS risk [28,31,33,34,35,36], whereas four did not [29,30,32,37]. Among the studies that did not find an association, two found slightly elevated risk estimates, especially for severe or repeated sexual abuse [30,37] but the associations were not significant. Although all the studies included ACEs, the definition and number of ACEs differed substantially. The most studied ACE was abuse, but the types and assessments of abuse varied. The cohort study from the Nurses’ Health Study investigated physical and sexual abuse, but did not include emotional abuse [37]. The largest case-control study combined physical and verbal abuse into one abuse exposure but did not include sexual abuse [29]. These two studies did not find associations between childhood abuse and MS. Interestingly, emotional and sexual abuse were the most common abuse categories associated with MS in the studies that reported associations [31,36,38].

All studies that investigated abuse used self-reported measurements of abuse through questionnaire screening tools, except for the one that used a computer-assisted telephone interview [29]. Only one study investigated early stressors assessed by national registry data [35]. Among the four studies with >100 participants that used registry data or validated abuse questionnaires, such as the Childhood Trauma Questionnaire [31,36] or Adverse Childhood Experiences Questionnaire [30], all except one found an increased MS risk [30]. In the studies that reported significant results, the risk varied from an 11% increase in MS risk to 18 times higher odds of MS after ACEs.

Recall bias is a potential challenge when assessing negative events occurring during childhood and may contribute to inconsistencies between studies. Childhood abuse tends to be underreported rather than overreported [39]. Only two cohort studies were not influenced by recall bias. This includes the Danish cohort study based on national registry data with deaths of household members and parental separation and the Norwegian cohort study that measured self-reported childhood abuse a median of 13 years before MS diagnosis. Both studies found associations between ACEs and MS development.

Evidence of an association between ACEs and MS susceptibility is supported by findings from experimental animal models of MS documenting that early-life stressors influence the susceptibility and severity of MS-like disease in exposed mice [20,22,23]. The combined evidence from experimental and epidemiological studies suggests that childhood adverse events contribute to a subsequent increased risk of MS.

## 4. Potential Mechanisms Underlying ACE-Related MS Risk

ACEs are associated with a type of stress often referred to as toxic stress, which comprises prolonged and excessive activation of biological stress response systems [40]. Toxic stress occurs in relation to chronic, recurrent, or severe early-life stressors, and when adequate adult support is lacking. Figure 1 illustrates how ACEs and toxic stress may affect adult health outcomes through dysregulation of pathways important for brain development, immune response systems, and epigenetic modifications. Toxic stress may therefore affect MS susceptibility through several direct and indirect pathways.

### 4.1. Neurodevelopment and Brain Structure

The development of the nervous system is particularly vulnerable to external stimuli during periods in childhood and early adulthood [41]. Different brain regions and neuronal pathways are sensitive to childhood abuse at different ages [42], and a range of associated molecular mechanisms may alter brain structure [41]. Ultimately, such alterations may increase the risk of neurological and neuropsychiatric disorders. Early-life stress is associated with increased permeability of the blood–brain barrier (BBB) [43]. Breakage of the BBB is also central in MS pathobiology and can occur throughout the course of the disease. Leakage through the BBB occurs even before any immune cell infiltration into the CNS and before myelin damage in MS [44,45,46]. BBB dysfunction in MS is thought to be caused by external factors, such as inflammation, but evidence suggests that alterations in the BBB are also caused by independent intrinsic mechanisms [47].

### 4.2. Dysregulation of Stress Responses and the Immune System

Abuse and trauma can lead to chronic activation of the hypothalamic–pituitary–adrenal (HPA) axis and inflammation [48]. Such inflammation could be a key mechanism linking ACEs to adverse adult health. Several studies have found that ACEs are associated with elevated concentrations of pro-inflammatory markers in peripheral blood that persist into adulthood [49,50] such as C-reactive protein, interleukin-6, and tumor necrosis factor alpha. Inflammation is a central part of MS pathobiology [51]. A dysregulated HPA axis is seen in MS [52]. High levels of the stress hormone cortisol are seen in over 50% of MS patients, and the immune cells of MS patients have higher resistance to glucocorticoids than immune cells in the controls, resulting in disrupted regulatory glucocorticoid response [52]. A dysfunctional stress system influences the immune system, and these interactions are relevant to MS pathogenesis.

### 4.3. The Neuroimmune Network Hypothesis

Childhood adversity not only influences individual organ systems but can also interact with the interplay between them. The neuroimmune network hypothesis claims that early-life adversity influences the crosstalk between peripheral inflammation and neural networks involved in the processes that make up the behavioral response to threat, reward, and control [53]. This disturbed crosstalk results in low-grade inflammation that contributes to a “pre-disease state”. The inflammation also affects the brain, where inflammatory cytokines alter cortico-amygdala threat circuits and cortico-basal ganglia reward circuits in a way that is believed to predispose individuals to risky health behaviors, such as smoking and high-fat diet consumption [53].

### 4.4. EBV

Emotional stress can reactivate herpes viruses, such as EBV [54]. Over 90% of the adult population is infected with EBV by their mid-20s. The virus appears in a latent resting state in immunocompetent individuals [55]. Early-life stress with exposure to ACEs is associated with high EBV antibody titers as an adult [56,57,58]. High EBV antibody titers are also associated with an increased MS risk [59,60]. EBV is believed to be a causal risk factor for MS [1], and genetic variants associated with reduced control of the EBV infection increase the MS risk [61]. It is unknown whether exposure to ACEs influences the immune response to the primary infection of EBV.

### 4.5. Genetics and Epigenetics

An individual’s risk of developing stress-related disorders after ACEs depends, in part, on genetic susceptibility. Epigenetic modifications have been seen in people exposed to childhood abuse, with altered expression of stress-related genes and increased risk of posttraumatic stress disorder (PTSD) in adulthood [62]. Stress disorders, such as PTSD, are associated with an increased risk of MS [63,64]

There are well-known interactions between the main environmental risk factors for MS, such as EBV, smoking, adolescent overweight, vitamin D, and sun exposure, and the main genetic risk variant for MS; HLA-DRB*15:01 [65,66,67]. This means that the risk for MS increases substantially if exposed to both an environmental risk factor and the genetic variant HLA-DRB*15:01 than it does for each factor separately. The main mechanism by which environmental factors interact with genetic risk variants is epigenetic changes [68]. Epigenetic changes are seen in both the brain and immune cells in people with MS [69]. Studies have yet to investigate interactions between ACEs and genetic risk factors for MS, and epigenetic changes in people with MS who have been exposed to ACEs.

### 4.6. Shortening of Telomere Lengths

Telomeres represent protective caps at the end of all chromosomes. They shorten as the cell ages. Adults exposed to ACEs have an accelerated shortening of telomere lengths [70]. People with MS show the same accelerated telomere shortening [71]. Environmental MS risk factors, such as obesity and smoking, contribute to telomere shortening [72]. Accelerated telomere shortening is mediated by chronic inflammation and oxidative stress, which, in turn, may cause genetic instability and dysfunction of immune cells [72]. Telomere shortening may be one mechanism for ACE-induced MS risk.

### 4.7. Behavior and Lifestyle

ACEs are associated with a range of behavioral and lifestyle factors that are also associated with increased MS risk, such as smoking, high body mass index, adverse socioeconomic status, and physical inactivity [4,25,73,74,75]. This highlights the importance of taking these well-known risk factors into account when examining the impact of ACEs on MS. Such factors may fully or partially mediate the association between ACEs and MS. Thus, it is crucial to include such variables in regression models to assess the direct effect of ACEs on MS risk. Of the 10 epidemiological papers included in this review, only four adjusted for ≥1 socioeconomic variable (income, education, deprivation) [28,30,31,36], four adjusted for smoking [28,30,31,37], and two adjusted for both smoking and body weight [28,37] Only our study adjusted for both socioeconomic factors, smoking and overweight [28], supporting an increased risk of MS after childhood abuse.

## 5. Childhood Adversity: Moderating Factors

Social support is regarded as the strongest moderating factor on the impact of stress [76]. The epidemiological studies on ACEs and MS have no information on protective aspects of childhood adversity, such as interventions or social support by caregivers or the community. Individual differences and resilience are relevant, as the reaction to similar stressors varies between individuals. Resilience is affected by behavioral and emotional coping mechanisms, genetic factors, and coexisting burdens, such as physical or psychological diseases. Furthermore, the reaction to stressors varies among individuals at different time points. Severity, chronicity, co-occurrence, and accumulation of childhood stressors all influence the risk of adult disease. We found a dose–response relationship between MS risk and the number of abuse exposures, which we interpreted as a measure of ACE severity [28]. Two studies distinguished between exposure in childhood (0–10 years) and adolescence (11–17 years [37] and 11–20 years [29]) but did not report significant associations between ACEs and MS risk for any of the groups. None of the studies reported the chronicity or duration of the ACEs.

Childhood maltreatment and abuse increase the risk of unfavorable coping with stress [77] and increase stress reactivity [78]. Thus, ACEs not only cause traumatic and long-lasting stress by themselves but also negatively affect the response to a wide variety of stressors in adult life. Furthermore, exposure to abuse and neglect in childhood increases the risk of experiencing abuse as an adult [79]. A Norwegian cohort study found a two-fold higher risk of experiencing repeated abuse as an adult after experiencing childhood abuse among women with MS compared to women without MS [38]. People with MS and a history of ACEs can therefore be exposed to a vicious chain of repeated stressful events and unfavorable coping. Stress and adverse coping are associated with increased MS disease activity, thus further increasing the consequences of ACEs on MS outcomes [12,32].

## 6. Limitations

The literature search was not performed as a systematic search, and some relevant literature may not have been included. In particular, the search terms may not have been adequate for detecting all relevant experimental studies on early-life stress and susceptibility to MS-like diseases.

## 7. Recommendations for Future Research

The epidemiological studies investigating the effects of childhood stress on MS are heterogeneous. Early stressors examined varied from parental separation and death to bullying and abuse. Most studies focused on abuse exposures, but they had different definitions of abuse. The studies did not clarify the role of the type of adversity, the synergistic effects between multiple forms of adverse events, the relevance of duration and timing of the ACEs, or the role of moderating and protective factors. The heterogeneity highlights the difficulty of measuring stress consistently and objectively within and across studies, as the types of stress and combinations of stressors give a wide range of possible exposures. One important aspect in increasing the knowledge and replicating findings on childhood adversity in MS would be to use consistent measures of ACEs across studies, such as the ACE International Questionnaire developed by the World Health Organization [80]. The ACE International Questionnaire includes the main types of abuse, neglect, household dysfunction, parental death and separation, bullying, and war trauma. Prospective cohort studies are preferable, as case-control studies are prone to recall bias.

Further studies on the underlying mechanisms of the association between ACEs and MS are warranted. Studies that examine interactions between genetic risk alleles and ACEs would provide important knowledge on gene–environment interactions. Interaction studies on other environmental risk factors and ACEs, such as markers of EBV infection, would provide insight into EBV-related mechanisms for MS susceptibility.

Biomarker panels to screen for toxic stress and to identify those vulnerable and those resilient to the negative effects of ACEs are in development [81]. A new and promising biomarker is the urokinase plasminogen activator receptor (suPAR), a marker of chronic inflammation [82]. A large prospective cohort showed that plasma suPAR had a stronger and more specific association with ACEs compared to traditional inflammatory markers, such as C-reactive protein and interleukin-6 [83].

## 8. Conclusions

Epidemiological research supports an association between childhood stressors and MS development despite some heterogeneity and limitations of published studies. An association between early-life stress and MS susceptibility has also been found in animal models for MS.

ACEs influence multiple biological pathways that are relevant to MS. A toxic stress response contributes to altered function of the neuroendocrine and neuroimmune systems, and to genetic and epigenetic changes that increase susceptibility to MS. Several pathways have interactive effects, such as the neuroimmune network, which regulates both peripheral inflammation and inflammation in the brain. In addition, ACEs increase the risk of an unfavorable lifestyle and exposure to other environmental risk factors for MS, such as smoking and being overweight. Further epidemiological studies applying consistent ACE measures combined with studies on the underlying mechanisms are needed to gain insight and enable the protection of children and adolescents at risk of developing MS.

## Figures and Tables

**Figure 1 ijms-25-01520-f001:**
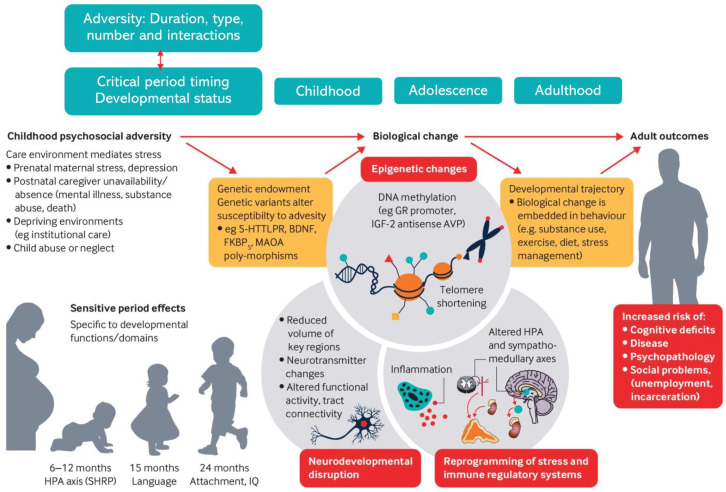
An illustration of potential pathways through which adverse childhood experiences (ACEs) can interact with genetic susceptibility and behavioral trajectories. ACEs induce toxic stress responses, which cause disruptions in neurological, endocrine, immune, and metabolic pathways and alter genetic function and expression. These changes increase the risk of adverse health and disease throughout the lifespan. The same pathways increase MS susceptibility. Reproduced in line with Creative Commons CC-BY-NC 4.0 license. ©2020 by British Medical Journal Publishing Group [27].

**Table 1 ijms-25-01520-t001:** Epidemiological studies investigating associations between adverse childhood experiences (ACEs) and MS.

Author (Year)	Study Design	Location of Study	Number of Participants	Type of ACEs	Childhood Period	Mean Age When Exposure Was Assessed	Mode of ACE Assessment	Results	Confounder or Mediator Adjustment
Eid et al. (2022) [28]	Prospective cohort	Norway, population-based	77,997 females 14,477 exposed to childhood abuse 63,520 unexposed to childhood abuse	Emotional, sexual, and physical abuse	0–18 years	29 years exposed 30 years unexposed	Self-reported questionnaire	300 developed MS during follow up. Emotional abuse (HR 1.40, CI 1.03–1.90) and sexual abuse (HR 1.65, CI 0.83–2.06) were associated with MS development. Physical abuse (HR 1.31, CI 0.83–2.06) Dose–response relationship: One category (HR 1.1, CI 0.79–1.56), Two categories (HR 1.66, 1.04–2.67), Three categories (HR 1.93, CI 1.02–3.67)	Birth year, school drop-out (≤9 years of elementary school), low household income, non-cohabiting parent, smoking, overweight
Horton et al. (2022) [29]	Case-control	Northern California	2607 participants 1422 MS (298 male, 1124 female) 1185 controls (219 male, 966 female)	Parental death, remarriage, divorce, and severe illness. Physical or verbal abuse, or neglect. Adopted. Loss of home, victim of violent crime.	0–10 years and 11–20 years	49.7 years	Computed assisted telephone interview (CATI) with 9 ACE categories	No significant association between ACE and the risk of MS (OR 1.01, CI 0.87–1.18)	Sex, birth year, ethnicity
Gatto et al. (2022) [30]	Cross-sectional	Iceland, population-based	28,870 females 214 MS	Physical, emotional, and sexual abuse, physical and emotional neglect, household dysfunction, community violence, and bullying	0–18 years	44.9 years	ACE-international questionnaire with 13 ACE categories Web-based survey	No significant associations between ACEs and risk of MS. Prevalence ratios were higher for physical neglect (PR 1.32, CI 0.71–2.32), parental separation/divorce (PR 1.21, CI 0.86–1.68), and bullying (PR 1.27, CI 0.83–1.91)	Age, education, childhood deprivation, smoking, depression
Wan et al. (2022) [31]	Case-control	Canada, Manitoba	681 participants MS: 232 Inflammatory Bowel Disease: 216 Rheumatoid Arthritis: 130 Healthy controls: 103	Emotional, physical, and sexual abuse, physical and emotional neglect	Age not specified “when I was growing up”	53.6 years	Childhood Trauma Questionnaire—Short Form with 28 items	The prevalence of having ≥1 maltreatment was higher in immune-mediated disorders than in controls (MS, 63.8%; IBD, 61.6%; RA, 62.3%; healthy controls, 45.6%). The trauma scores were also higher for all types of abuse and neglect People with immune-mediated disorders had an OR of 2.37 (CI 1.15–4.89) for emotional abuse	Age, sex, ethnicity, smoking status, years of formal education, and annual household income
Briones-Buixassa et al. (2019) [32]	Case-control	Spain	41 MS 41 controlsFemale:Male ratio 70:30	Emotional, physical, and sexual abuse, physical and emotional neglect	Age not specified“when I was growing up”	48.5 years for MS 48.0 years for controls	Childhood Trauma Questionnaire- Short Form with 28-items	No significant association between early-life stress in MS (*p* = 0.65) People with MS had a higher mean score of emotional abuse and neglect than controls, but not significant (*t*-test, *p* = 0.08)	None
Shaw et al. (2017) [33]	Cross-sectional	New York	67 MS 15 males 52 females No control group	Emotional, physical, and sexual abuse. Household dysfunction and neglect	0–18 years	50.5 years	ACE-questionnaire with 10-items	Increased occurrence of experiencing >4 ACE compared to the Kaiser Permanente ACE-study	None
Eftekharian et al. (2016) [34]	Case-control	Hamadan, Iran	250 MS 64 males 186 females 250 controls 69 males 181 females	Physical child abuse, head trauma, stress and anxiety disorders, OCD, depression	Age not specified	Not reported	Interview with questionnaire	People with MS had increased risk of childhood physical abuse 2–3 times/week (OR 18.81, CI 4.46–79.38)	None
Nielsen et al. (2014) [35]	Retrospective cohort	Denmark, population based	2.9 million participants 3260 MS cases	Parental divorce, parental death, death of sibling	0–18 years	n/a	Stressful life events obtained from the Danish Civil Registration System	Exposure to one stressful event gave increased risk of MS (RR 1.11, CI 1.03–1.20) Parental divorce gave 13% increased risk for MS (RR 1.13, CI 1.04–1.23). Exposure to parental or sibling death did not increase MS risk	Sex, age, calendar period
Spitzer et al. (2012) [36]	Case-control	Germany	1119 participants 234 MS170 females, 64 males 885 controls	Emotional, sexual, and physical abuse and emotional and physical neglect	Age not specified“when I was growing up”	39.7 years for MS 41.2 years for controls	Childhood Trauma Questionnaire- Short Form with 28 items	Higher risk of emotional abuse (OR 3.4, CI 2.0–5.7), sexual abuse (OR 2.2, CI 1.1–4.2), and emotional neglect (OR 2.0, CI 1.2–3.2), among people with MS. The odds of physical abuse were 1.3 (CI 0.6–2.6) and 1.5 (CI 0.9–2.4) for physical neglect	Age, sex, educational level, and current depression
Riise et al. (2011) [37]	Mixed cohort Mainly retrospective	United States, Nurses’ Health Study	116,671 female participants 292 MS 49 MS individuals followed prospectively	Sexual and physical abuse	Childhood 0–10 years Adolescence 11–17 years	Not reported	Self-reported questionnaire	No significant associations. Elevated estimates for being repeatedly forced into sexual activity in childhood (OR 1.47, CI 0.87–2.48) or adolescence (OR 1.21, CI 0.68–2.17)	Age, ethnicity, latitude gradient, body mass index, smoking

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
