# Peer review of "Adverse Childhood Experiences and the Risk of Multiple Sclerosis Development: A Review of Potential Mechanisms"

_ijms, 2024, doi:10.3390/ijms25031520_

Round 1

Reviewer 1 Report

Comments and Suggestions for Authors

The authors reviewed evidence on the association between ACEs and adult-onset MS. Recommended changes are listed below. 

1. Please provide a summary of the epidemiological studies in addition to the table. This paragraph would make more sense right next to the table.

2. In the Cohort studies section, the last paragraph is written in first person - was this study conducted by all the current authors? 

3. With two recent systematic reviews on the topic - it is not clear why this review is strongly justified.

4. A challenge throughout this review is that ACEs are differently defined across the studies reviewed. Given that different ACEs are thought to lead to different neurobiological and psychological outcomes, this is an important potential confounder to consider. 

5. The neuro-immune network hypothesis would be an important model to discuss in the introduction and discussion.

6. Stating the number of studies included in each section of the review would be helpful for interpreting the results. For example, in the epi section, it is noted that all except one found an increased risk, but one of how many? Also, given the different definitions of ACEs - did this impact outcomes differently?

7. Please provide a summary of all the literature reviewed before moving to mechanisms. Given that some studies found no association, it is important to discuss the discrepancies and reasons these studies may not have found the association. 

8. Given that this is a review paper, occasionally referencing the authors' previous study using the first person is rather confusing. 

9. The conclusion that epi and experimental research supports an association is somewhat weak given that there were only 2 mouse studies reported in the experimental section and there was one epi study with null findings. This section should provide a more detailed and nuanced overview of the literature.

Comments on the Quality of English Language

The manuscript was generally well-written, although use of terms such as "probably" undermine the conclusions. Further, a minor issue, use of the word "this" without a clear antecedent is confusing in multiple places. 

Reviewer 2 Report

Comments and Suggestions for Authors

This paper is overall very well-written and covers a highly interesting and relevant topic. Below, I list my comments by section and/or line number.

Introduction

Line 28: The authors state that EBV infection is thought to be a "causal" risk factor for MS. I would stress that EBV is a "putative causal" or "possibly causal". The study cited states "Evidence of causality remains inconclusive."

Line 29: "increasing the risk of MS 32-fold": Again, there is a very strong association, but I would hesitate to state that EBV increases the risk. EBV is *associated* with an increased risk.

Line 30: "Studies have found that EBV infection in adolescence gives a particularly increased risk...". As mentioned previously, please be careful not to over-state this observation/association.

Line 38: "...when Jean-Martin Charcot described 'long-continued grief or vexation' to be assigned causes of the disease." Charcot observed that grief or vexation were related to disease onset but did not go so far as to assign causes of the disease.

Line 65: The semicolon (;) should be a colon (:).

Search strategy

Lines 76-77: The search terms used were acceptable; however, as discussed below, please indicate that this search was by no means exhaustive, and some relevant literature may not have been included.

Line 81: The authors state that papers were excluded in the full text was not available. However, please include more details regarding how you defined "unavailable." Did this mean the papers were not available for access from the authors' respective institutions? Were the papers not available in English (or another language)? 

Adverse childhood experiences are associated with increased MS risk

Line 88: Please define why you say there is a "growing interest." For example, did you notice an increased pattern of number of publications or citations?

Line 89 and Table 1: Is there a legend for this table, or explanation about why these specific studies were included in the table? I see you have written "An overview ...is shown in table 1" in line 89, but the studies included in the table look more like a subjective cherry-picking than an overview. Perhaps it would be better to say the table includes examples highlighted in the text. This will help the reader understand why these specific studies were included.

Experimental studies

As mentioned previously (regarding the Search strategy terms), some relevant studies were not included in this review. I suggest the authors either add more search strings to the PubMed search, or leave out this section altogether, because it is certainly not adequate as it currently is written. This is not an exhaustive list of "experimental studies". For example, why did the authors neglect to cite other experimental models of MS, such as the Theiler's murine encephalomyelitis virus model, which has also been used to evaluate the effects of early life stress?

Case-control studies

Line 109: "An association between ACEs and MS was first reported in a German case-control study..." Please remove the word "first".

Line 115: "with an unadjusted odds ratio of 18.8" There is a period (.) missing.

Lines 128-129: "...assessed early life stressors such as parental divorce and death of parents or siblings from national registries." The way this is worded makes it sound like their deaths were caused by national registries. (I know this is not at all what you meant!) Please add commas (,) or otherwise restructure the sentence for clarification purposes.

Lines 135-136: "The study was mainly retrospective, but followed 49 women who developed MS prospectively." I was left confused by this statement. Do you mean to say that the retrospective study included 49 women who eventually developed MS and who were included in a follow-up prospective study?

Neurodevelopment and brain structure

Line 206: "...alterations in the BBB is also caused..." This should read as "...alterations in the BBB are also caused..."

In conclusion, I found this review to be interesting, though perhaps not as comprehensive as the authors would like. Please include a section for "Limitations" or something like that so the reader understands the review is not comprehensive. (Most reviews are not! But please let your reader know you are aware of what your review truly covers, and what it does not cover.)

Comments on the Quality of English Language

The English language is of high quality. I have flagged only a few minor instances of punctuation or grammar errors.

Round 2

Reviewer 1 Report

Comments and Suggestions for Authors

Thank you for your revision - the authors have addressed my concerns.

Reviewer 2 Report

Comments and Suggestions for Authors

The authors are commended for their thorough and conscientious responses to reviewer comments.